# Geo-Spatial Characteristics of 567 Places of Tick-Borne Encephalitis Infection in Southern Germany, 2018–2020

**DOI:** 10.3390/microorganisms10030643

**Published:** 2022-03-17

**Authors:** Amelie M. Friedsam, Oliver J. Brady, Antonia Pilic, Gerhard Dobler, Wiebke Hellenbrand, Teresa M. Nygren

**Affiliations:** 1Immunization Unit (FG33), Robert Koch Institute, Seestraße 10, 13353 Berlin, Germany; pilica@rki.de (A.P.); wkhellenbrand@gmail.com (W.H.); nygrent@rki.de (T.M.N.); 2Centre of Mathematical Modelling for Infectious Diseases, London School of Hygiene and Tropical Medicine, London WC1E 7HT, UK; oliver.brady@lshtm.ac.uk; 3Department of Infectious Disease Epidemiology, Faculty of Epidemiology and Population Health, London School of Hygiene & Tropical Medicine, London WC1E 7HT, UK; 4Department of Microbiology of the German Armed Forces, 80937 Munich, Germany; gerharddobler@bundeswehr.org

**Keywords:** tick-borne encephalitis, spatial epidemiology, risk mapping, ecological niche modelling, Germany, Bavaria, Baden-Wuerttemberg

## Abstract

Tick-borne encephalitis (TBE) is a growing public health problem with increasing incidence and expanding risk areas. Improved prevention requires better understanding of the spatial distribution and ecological determinants of TBE transmission. However, a TBE risk map at sub-district level is still missing for Germany. We investigated the distribution and geo-spatial characteristics of 567 self-reported places of probable TBE infection (POI) from 359 cases notified in 2018–2020 in the study area of Bavaria and Baden-Wuerttemberg, compared to 41 confirmed TBE foci and 1701 random comparator places. We built an ecological niche model to interpolate TBE risk to the entire study area. POI were distributed heterogeneously at sub-district level, as predicted probabilities varied markedly across regions (range 0–93%). POI were spatially associated with abiotic, biotic, and anthropogenic geo-spatial characteristics, including summer precipitation, population density, and annual frost days. The model performed with 69% sensitivity and 63% specificity at an optimised probability threshold (0.28) and an area under the curve of 0.73. We observed high predictive probabilities in small-scale areas, consistent with the known circulation of the TBE virus in spatially restricted microfoci. Supported by further field work, our findings may help identify new TBE foci. Our fine-grained risk map could supplement targeted prevention in risk areas.

## 1. Introduction

Tick-borne encephalitis (TBE) is the most widespread vector-borne disease in Europe [1]. TBE incidence has been increasing during the past decades and represents a growing public health threat in many endemic countries [2,3]. The TBE virus is a flavivirus that can affect the central nervous system of infected individuals [1]. While an estimated 70% to 95% of infections progress without or only with mild symptoms, approximately one-third of symptomatic cases experience symptoms of the central nervous system, such as meningitis or meningoencephalitis; the overall mortality is 1% [4]. The Western TBE subtype endemic in Europe is predominantly transmitted by *Ixodes ricinus* ticks [5].

Between 2012 and 2016, 12,500 cases were notified in the European Union and European Economic Area, with the annual case number fluctuating around 2500 [6]. In Germany, about 300 TBE cases are notified each year [7], most (85%) from the southern federal states of Bavaria and Baden-Wuerttemberg [7]. Between 2001 and 2018, TBE incidence showed an annual increasing trend of 2% [3]. In 2020, the highest case number since the start of data collection in 2001 was reported, at 712 TBE cases, i.e., 60% higher than the previous year [7,8]. While the underlying reasons for this development are manifold, it is suspected that more people were exposed to tick bites because of increased time spent in nature and forests during the COVID-19 lockdown and international travel restrictions [8,9,10].

The three prerequisites for TBE presence and persistence are the vector, virus, and host animals [11]. All these components are sensitive to a variety of meteorological and ecological factors [11,12,13,14,15,16,17], which is why infection risk often concentrates in highly localised foci that are characterised by suitable conditions for tick and host survival, virus replication, and tick questing behaviour [18,19]. Tick activity and consequently TBE transmission show pronounced seasonality [7]. Tick larvae and nymphs generally start feeding in spring, reach peak activity in summer, and stop questing in autumn [13,20]. Most TBE transmission in Germany occurs between May and October [7]. TBE has been repeatedly reported in new locations within Europe and Germany, implying that endemic areas have increased and that the geographic distribution of the virus has expanded [21,22,23]. It is therefore crucial to not only understand the ecological conditions that determine the ticks’ survival, activity, and spatial distribution, but also the conditions which permit establishment of TBE foci.

Fructification can predict rodent numbers, which are often related to TBE occurrence [24]. Previous research has used the beech fructification index as a main predictor for the TBE virus transmission cycle to generate TBE incidence forecasts for Germany, but did not produce a risk map [24]. A map generated by Brugger et al. of the density of *I. ricinus* in Germany only represents the vector as one determinant of TBE risk but does not account for anthropogenic aspects of human TBE infection such as population density or vaccination coverage [25]. The spatial pattern of TBE risk in Germany has only been mapped at the district level due to routine surveillance data being collected by district [7,26]. The results from this spatial analysis will enable a more detailed perspective on a smaller scale because TBE risk is probably not evenly distributed across districts [3].

This study aims to (i) assess the spatial distribution and clustering of self-reported places of infection in Bavaria and Baden-Wuerttemberg between 2018–2020; (ii) describe selected characteristics of places of infection and compare them with those of randomly generated comparator places and known natural TBE foci, i.e., precisely characterised areas with confirmed TBE virus prevalence in ticks; and (iii) build a multivariable ecological niche model and generate a predictive map to understand the importance of environmental factors in determining the likelihood for TBE transmission and to identify new at-risk areas or potential areas of underreporting.

## 2. Materials and Methods

### 2.1. Places of Infection

All TBE cases from the study area of Bavaria and Baden-Wuerttemberg notified from 1 January 2018 to 31 December 2020 were invited to participate in a study. Cases were recruited on an ongoing basis throughout the study period and were interviewed as soon as possible to minimise recall bias. Full details on the study design are published separately [10]. Cases reported probable places of TBE infection (POI), i.e., places where a tick bite could have occurred within the exposure period of four weeks prior to symptom onset. Cases marked one or more probable POI as points or areas on printed A4 maps scaled to cover the respective administrative districts. Cases also reported the number of noticed tick bites within the exposure period (categories as “no tick bite” or “one or more tick bites”). Three POI without information on the occurrence of tick bites were excluded. The overall study included 581 participants out of 1220 eligible TBE cases in Germany (48%). Sixty-two percent of these provided information on the probable place of TBE infection located within the defined study area [10]. POI were georeferenced as points or polygons in QGIS 3.18 [27]. To account for inaccuracies in the case markings on the map, buffer zones were generated around each point, ranging between 300–400 m depending on the printed map’s scale. Large polygons were not considered sufficiently specific for the purposes of this analysis; hence, we excluded polygons larger than the median size of all polygons at 5.4 km^2^ (247 of 494 polygons excluded).

### 2.2. Confirmed Natural TBE Foci

TBE foci are small geographic areas with virus-carrying ticks and suitable conditions for TBE transmission [28,29,30]. We used a set of 41 natural TBE foci which were initially reported by TBE cases and subsequently identified by experts as places where TBE virus-positive ticks were collected during field work based on previously described methods [28,29,30]. Together, natural foci and POI make up the TBE presence data for this analysis.

### 2.3. Comparator Places

Comparator places were randomly generated within the study area in QGIS with a 3:1 ratio of the POI to ensure sufficient statistical power. Comparator points were generated with the same buffer zone as the POI points. Comparator polygons were generated with the mean size of the POI polygons included in the analysis. An altitudinal limit of up to 1000 m ensured that the centroids of the comparator places were not generated above a height where ticks are very rarely or never present [31]. Only one place from our TBE dataset was located above 1000 m.

Relevant abiotic, biotic, and anthropogenic parameters for TBE endemicity and tick ecology were identified from the literature. PubMed was searched for the terms “TBE” OR “tick-borne encephalitis” OR “Ixodes Ricinus” AND “risk” OR “ecolog*” OR “environment*” OR “geograph*” OR “habitat” OR “climat*”. Additional articles were selected through snowball sampling [2,6,12,16,32,33] and expert recommendations [10,30,34].

### 2.4. Covariates

Based on existing research, summer and spring temperature, summer precipitation, land cover, snow cover days, frost days, and hot days were included as abiotic determinants into this analysis [1,12,13,18,19,20,35,36,37,38]. We moreover included important anthropogenic and biotic factors such as vaccination coverage, population density, and tick density [25,39,40].

#### 2.4.1. Abiotic Factors

The Digital Terrain Model (DGM200) containing altitude data from the Bundesamt fuer Kartografie und Geodaesie (BKG) describes the terrain forms of Germany’s surface by a set of points arranged in a regular raster at a 200 m resolution with a ~3–10 m height accuracy depending on the terrain type [41]. Land cover data also came from the BKG [42]. The latest version (2018) of the German digital land cover model (LBM-DE2018) describes the topographic characteristics of the landscape in a polygon format. There were 31 land cover types ranging between artificial surfaces, agricultural areas, natural areas, and water bodies [42].

Meteorological data were accessed through the German Weather Service (DWD) [43]. Raster files with a 1 × 1 km resolution were obtained for each study year for temperature indicators, precipitation, and snow cover. A seasonal raster of the monthly averaged daily summer temperature measured two meters above the ground (hereafter “mean summer temperature”) was obtained [43]. In addition, temperature extremes were included through the number of hot days (>30 °C) and the number of frost days (<0 °C) per year [43]. Raster files for the seasonal sum of summer precipitation height (mm) and the annual number of snow cover days (≥1 cm) were downloaded from the DWD [43].

#### 2.4.2. Biotic and Anthropogenic Factors

A raster file for the modelled density of *I. ricinus* in Germany was provided by Brugger et al., indicating the number of ticks per 100 m^2^ [25]. A polygon file for population density at the community level was obtained from the BKG for 2018 [40]. Lastly, coverage of complete and timely vaccination per district population (%) for 2018 was obtained through the national vaccination monitoring system “KV-Impfsurveillance” of the Robert Koch Institute (RKI) [39].

### 2.5. Spatial Analysis

Each POI was weighted according to the inverse of the number of places each participant reported (median = 1; range 1–18). For instance, if a participant marked 5 places, each place was weighted with 1/5. The rationale for these weights was a probable lack of certainty and specificity for each individual place, if participants marked multiple places. A choropleth map for the aggregated count of weighted POI per district helped visually investigate spatial heterogeneity. Furthermore, a Kernel Density Estimation (KDE) heatmap was created with a 10 km bandwidth to examine whether there was spatial heterogeneity in the distribution of weighted POI at sub-district level.

QGIS was utilised to visualise and process covariates. Through sampling zonal statistics in QGIS, the mean covariate raster values of the three types of places were determined. Boxplots were created in R 4.0.5 [44] to compare the three types of places in their covariate characteristics. Furthermore, the proportion of each land cover type per the sum area of POI, comparator places, and TBE foci was calculated.

A multivariable ecological niche model was built in R using a generalised linear model (GLM) approach, with the POI and confirmed foci as presence data and the random comparator places as pseudo absences. The model was used to assess the importance of the covariates and interpolate the probability of POI presence to the entire study area. Raster files of all covariates were resampled to the same resolution of 1 × 1 km and stacked. Data from 2019 were selected as a reference for the weather covariates, after inspecting the time series data of the weather covariates throughout the study period. Prior to building the model, covariates were examined for multicollinearity through the Variable Inflation Factor (VIF). In a stepwise selection, the covariates with the highest VIF scores were removed iteratively until all covariates had a VIF < 10, which is a commonly used threshold [45]. Based on the most prominent land cover types of TBE foci, land cover data were rasterised and dichotomised into presence or absence of forest. The model was initially run with all nine covariates. Automated stepwise backward selection of covariates by Akaike’s Information Criterion (AIC) was then performed. The model with the lowest AIC value was selected as the final model [46]. Predictive model performance was evaluated by inspecting the receiver operating characteristic (ROC) curve and calculating the area under the ROC curve (AUC) [47,48]. The magnitude and direction of effect of each covariate was assessed by comparing the model predicted risks with each covariate fixed to the 25th and 75th percentiles of their distribution, with all other covariates held constant at the median (50%) value [49]. This was then used to calculate an odds ratio (OR) between comparatively “low” and “high” values of each covariate that were approximately comparable between different covariates.

Finally, sensitivity analyses using presence data were restricted to the confirmed foci and only (a) POI indicated by participants who reported one or more tick bites (59% of all POI), (b) POI smaller than 1 km^2^ (66% of all POI), and (c) POI that were reported as a standalone place by cases (33% of all POI), to enhance the precision of the presence data. This also allowed for one to investigate potential mischaracterisation of the attributes of the true focus within the reported POI when these were reported as larger or alongside several other places. The sensitivity analyses models were compared with the ecological niche model to examine its performance and quality.

Based on the final ecological niche model, a continuous and a binary predictive risk map was created in QGIS and superimposed with POI and TBE foci to compare the presence data with the model predictions. The continuous map was further compared to maps created from sensitivity analyses, a Kernel Density Estimation heatmap for reported POI, and an excerpt of the official RKI TBE risk map [7,27].

## 3. Results

### 3.1. Epidemiological Overview and Descriptive Statistics

The analysis included 567 POI reported by 359 study participants (Table 1). The median delay between symptom onset and reception of POI maps marked by the cases was 83 days (interquartile range = 57–132 days). The size of all types of places in km^2^ is shown in Table 2 and their geographical location in Appendix A.

Although Baden-Wuerttemberg is only half the size of Bavaria, it accounted for 53% of the weighted POI. Of all 157 districts, Ravensburg had the highest sum of weighted POI with a count of 23, followed by Zollernalbkreis (weighted count = 17) (Figure 1A).

Temporal and spatial selection biases were investigated to examine the representativeness of the study data as compared to the overall TBE infection pattern. Examination of the spatial distribution data did not raise concerns about bias, as the distribution of the study data was comparable to all notified TBE cases in the study area (Appendix A).

### 3.2. Spatial Clustering of Places of TBE Infection

On visual inspection, the distribution of the POI appears spatially heterogeneous (Figure 1). Most places were reported in Baden-Wuerttemberg in the districts Ravensburg, Zollernalbkreis, Bodenseekreis, and Freudenstadt. In Bavaria, the districts with the highest weighted count of places were Rosenheim, Schwandorf, and Unterallgaeu. Only few places were concentrated in the northern and central areas of the study region. Figure 1B displays the same data but unaggregated as a heatmap, revealing clear contrasts in the density of POI within the district. For instance, the heatmap for Ravensburg (darkest district in Figure 1A) showed POI only concentrated in the southern region of the district.

### 3.3. Individual Environmental Characteristics of Places of TBE Infection

It is reasonable to assume that the specific suitability of TBE foci as tick habitats, as demonstrated by the observed presence of TBE virus-carrying ticks, may be reflected in distinctive environmental characteristics of these foci [29]. Therefore, characteristics of comparator points are expected to differ from those of foci, since comparator points were generated randomly. Knowledge of distinctive individual environmental characteristics may be informative for identifying new POI. Moreover, if environmental characteristics of POI were similar to those of the natural foci and different from comparator places, this may imply that the reliability of the POI is high. Selected abiotic covariates were thus compared between POI, TBE foci, and comparator places.

The value ranges of multiple covariates differed considerably between the three types of places (Figure 2). Partly driven by the small sample size, the value range of TBE foci was the smallest for most covariates (interquartile range and general spread).

The average number of “annual frost days”, “annual snow cover days”, and “mean summer temperature” barely differed between POI and comparator places. TBE foci had slightly more “frost days” and less “snow cover days” per year. Foci were on average 0.4 °C warmer during summer. None of the abiotic covariates showed strikingly similar characteristics between POI and TBE foci or striking differences from comparator places. Further descriptive statistics are summarised in Appendix A.

Land cover for all three types of places was predominantly characterised by natural surface such as forest and grassland, yet the proportional shares of the categories differed substantially (Figure 3). Natural foci were mainly made up by coniferous trees (59%), and POI by homogeneous grassland (23%). Strikingly, the largest proportion (29%) of comparator places was made up by arable land, contrasting the other two types of places. There was more diversity in land cover types at the large POI and comparator places than in the small TBE foci. For instance, arable land, houses, and grassland with trees were observed within POI and comparator places but not within TBE foci. Houses accounted for a small but considerable proportion (9%) of the POI. Although such artificial land cover types are usually not considered among the main risk areas for TBE, studies have found residential gardening to be a significant risk factor for tick bites [32,34].

### 3.4. Ecological Niche Model

Eight covariates remained in the final model after VIF exclusion and stepwise backwards selection (Figure 4). “Hot days per year” was the only covariate removed based on AIC changes. Figure 4 illustrates the direction and magnitude of effect of each covariate comparing model-predicted risk between the 75th and 25th percentile value of the covariate. Values greater than one suggest that higher values of that covariate increase the risk of POI presence. The greatest ORs for POI presence were found for “summer precipitation”, “annual frost days”, and “population density” (OR 2.8, 2.3, and 1.8, respectively). Increases in “tick density” and annual “snow cover days” were negatively associated with the probability of POI presence (OR 0.8 and 0.5, respectively). The predictive performance of the final ecological niche model was acceptable with an AUC of 0.73. Predictions were made with 69% sensitivity and 63% specificity at an optimised POI-presence probability threshold of 0.28, maximising the sum of sensitivity and specificity of the model.

#### 3.4.1. Predicted Probabilities for Places of TBE Infection

The predictive map based on the ecological niche model suggested high probabilities up to 93% for the presence of a POI in central, south-eastern, and south-western Baden-Wuerttemberg (Figure 5A). In Bavaria, high probabilities were predicted in the southern districts Lindau, Oberallgaeu, Wolfratshausen, and Berchtesgardener Land, in the centre of Munich and Neu-Ulm, and in the northern regions of Regensburg, Nuernberg, and Amberg (Figure 5A). Areas with high and low probabilities were spread heterogeneously across the study area. Except for some mismatches (many presence points vs. low predicted model probability) in the west of Baden-Wuerttemberg and (few presence points vs. high predicted model probability) in Oberallgaeu, the map accurately reflects the geographic distribution of the actual POI and TBE foci (Figure 5B). The binary risk map superimposed with POI and TBE foci further confirms this observation (Appendix A).

#### 3.4.2. Sensitivity Analysis and Kernel Density Estimation

Increased precision of the presence data was achieved by using different criteria for the inclusion of POI as model input: (a) only POI indicated by participants with a tick bite, (b) only POI smaller than 1 km^2^, and (c) only POI indicated as a standalone place by a case (Appendix A). Sensitivity analyses produced maps with smaller areas of high probability when compared to the final ecological niche model. This effect was anticipated since the sample sizes of presence data were reduced substantially (by 41%, 34%, and 67%, respectively) after applying the precision criteria for POI. Nonetheless, the overall spatial pattern and location of areas with high and low probabilities remained the same.

The ecological niche model (Figure 5A) was moreover compared to the Kernel Density heatmap based on the actual presence data (Figure 1B). Although two different outcomes are being portrayed in those maps, the similarity or dissimilarity of the spatial patterns may reflect the predictive performance of the model. The areas where the ecological niche model predicts the highest probabilities for the presence of a POI were only in some cases congruent with high densities in the heatmap. While the heatmap showed the highest density of weighted POI in two localised hotspots in Ravensburg, the ecological niche model only predicted high probabilities in the western hotspot. Another discrepancy between the two maps was observed in the south-western Bavarian district Oberallgaeu, where a low Kernel density compares to a high predicted probability of a POI being present. Based on visual inspection of the individual covariate rasters in this area (see selected raster maps in Appendix A), this discrepancy cannot markedly be attributed to one specific covariate but more likely to an interplay of multiple factors.

#### 3.4.3. RKI Risk Map and Ecological Niche Predictions

Most areas with high probabilities predicted by the ecological niche model (orange/red areas) seem to be in congruence with the areas of medium or high TBE incidence (≥2.65 to <52.42 cases/100,000 inhabitants/5 years), except for Munich, which is not marked as at risk in the RKI map (Figure 5C) [26]. The RKI map shows an elevated risk in the far east of Bavaria and some additional districts in the centre of the study area, which is however not reflected by the ecological niche model. For eastern Bavaria, the actual presence of data points seems to be in better agreement with the RKI map than the ecological niche model prediction.

## 4. Discussion

This analysis described environmental characteristics of self-reported POI between 2018–2020 and explored ecological niche modelling for interpolating the spatial distribution of reported places to all of Bavaria and Baden-Wuerttemberg, taking several potential explanatory factors into account. The choropleth map (Figure 1A) implies spatial heterogeneity in the places’ geographical distribution. Reported POI were predominantly clustered in central and southern Baden-Wuerttemberg and central-eastern and south-eastern Bavaria. The Kernel Density heatmap revealed clustering at the sub-district level (Figure 1B). This finding is in line with the previously described phenomenon of highly localised hotspots of TBE infection instead of more uniform distribution across larger areas [28].

There were only small absolute differences in environmental characteristics between TBE foci, self-reported POI, and comparator places. This implies that none of the tested abiotic covariates were distinctly specific for TBE risk when covariates were considered in isolation. This likely reflects that tick ecology is a multi-faceted interplay of multiple determinants [3,17]. It remains questionable to what extent the observed, limited level of variation can impact tick ecology.

The land cover proportions of TBE foci, and to a lesser extent POI, reflected previously published characteristics of a tick-friendly environment [35,36,37]. Forest and grassland were the predominant land cover types (Figure 3). Arable land has not been described as a typical tick habitat [50] but accounted for a considerable proportion of the POI. Similarly, observed artificial land cover types such as houses are not usually considered as an important tick habitat; however, gardening activities have been linked to an increased risk of tick bites [32]. Due to the relatively large sizes specified for some POI and thus comparator places, they were likely to include additional and sometimes artificial types of land cover by chance; thus, the true land cover attributes of the presumed TBE microfoci within the specified POI may not have been captured appropriately. By contrast, the confirmed TBE foci were smaller and thus likely to have a more precise and plausible land cover composition.

The multivariable ecological niche model provided additional insights into magnitude and direction of effect of the environmental parameters when considered in combination. The most important predictors for POI presence were “summer precipitation”, “population density”, and “annual frost days”, with the first two having understandable links with creating favourable tick habitats and increasing exposure of humans to tick bites. The surprising positive OR for “vaccination coverage” is likely explained by an association between a local awareness for an already existing TBE risk and a higher vaccination uptake in response to that. While frost and persistently low temperatures are known to inhibit tick questing and limit the chances for *I. ricinus* overwintering [12,38], our data suggested a positive association between “annual frost days” and POI presence (OR 2.3). The observed negative association between tick density and POI presence (OR 0.8) is not in line with ecological reasoning and should therefore be investigated more closely in the future. Another ambivalent direction of effect was observed for “annual snow cover days” (OR 0.5), which is generally considered as a protective factor for tick survival, as it prevents the ground temperature from dropping to unsuitably low temperatures [51]. On the contrary, suggesting non-linearity of this ecological covariate, extended snow cover periods may delay the beginning of spring in the environment, making habitats less favourable for *I. ricinus* and their hosts [51].

The model was further able to characterise the potential range of habitats that may be related to higher risk of TBE infection. It also confirmed the phenomenon of localised TBE hot spots and spatial dependency between POI presence and selected covariates. The predicted probabilities of the model were in good agreement with the presence data. Small mismatches between the prediction and the actual POI may be explained by either (a) under-reporting of TBE POI in some areas, (b) few people visiting certain areas prone to TBE transmission, (c) or low abundance of host animal populations.

Nonetheless, the sensitivity analysis map suggested a high quality of the self-reported data even when no tick bites were reported, since the overall pattern of the location of high- or low-probability areas was comparable (Appendix A). Sensitivity analysis with presence data restricted to more localised (<1 km^2^) (Appendix A) or standalone POI (Appendix A) resulted in a similar distribution of high and low probabilities, further confirming good quality of the POI dataset. High TBE incidence as defined by the RKI and a high predicted probability of the presence of a POI were moderately congruent (Figure 5B,C). In eastern Bavaria, the ecological niche model underpredicted the presence of POI compared to the RKI incidence map and the actual presence data. This suggests that the ecological niche model may require additional information to correctly classify these areas as potential POI.

### 4.1. Strengths

To the best of our knowledge, this is the first such comprehensive dataset on probable POI in Germany. The detailed, case-derived information allowed for spatial analysis on a smaller scale than before. The study data did not appear to express spatial or temporal bias when compared with the complete, routinely collected TBE surveillance dataset of notified TBE cases in Bavaria and Baden-Wuerttemberg, also including non-participating cases. Since all the information stems from the same study and was collected in the same way, findings were robust to varying definitions of self-reported POI. Furthermore, expert-derived high-precision data on previously characterized and confirmed natural foci helped assess the quality and informative value of self-reported POI.

### 4.2. Limitations and Recommendations for Future Research

Limitations include that (a) host animal abundance was not analysed as a potential predictor in the ecological niche model due to lack of data. In case that this data is not available for future analyses, incorporating spatially correlated random effects into the model may be a way to account for residual spatial unmeasured confounding. Furthermore, including the anthropogenicity of the factor “human outdoor activity” would enable a more comprehensive approach in determining potential POI. (b) The here-analysed macro-climatic covariates may not be precise enough to adequately explain and characterise the POI. Especially for highly localised vector-borne diseases such as TBE, it may be more important to incorporate microclimatic conditions with higher spatial resolution and with more detailed information on flora, fauna, and vegetation at the ground level. According to the literature, relative humidity was another relevant factor, but raster data were not available. Future research should further fine-tune the explanatory parameters to enhance the model’s predictive power. (c) Using generalised linear models in this context may not be ideal, as most abiotic covariates do not have a linear relationship with tick survival and TBE transmission [21]. Therefore, more advanced statistical methods such as changepoint models, additive models, or machine learning techniques accounting for non-linear interactions between variables are advised for future analyses. 

Future work may also be directed at the entirety of Germany to examine why TBE incidence is low in the north and higher in the south. A larger contrast in environmental characteristics may be seen along the north–south gradient, allowing better insight about why 85% of Germany’s TBE disease burden is localized in Bavaria and Baden-Wuerttemberg. More attention should also be paid to the sub-district distribution of POI to identify additional TBE foci.

Lastly, different algorithms used for ecological niche models and species distribution models can return different estimates [33]. Comparing multiple predicted models for the same study region may help discover areas of agreement regarding the suitability for TBE foci and the probability of disease transmission [33].

### 4.3. Public Health Implications

These results serve as an overview of the potential POI between 2018 and 2020 and may help guide surveillance, data collection, and public health planning in the future. Ideally, recommendations on public health measures should be formulated in comprehensive synergy alongside other existing information about TBE risk to discuss preventive strategies. More localised insights into high-infection densities at the sub-district level as seen for south Ravensburg (Figure 1B) may have relevant implications for improving targeted prevention measures such as providing the population with detailed information on local infection risk. This could encourage persons living in or visiting particularly high-risk areas to apply tick-protection measures and obtain TBE vaccination.

## 5. Conclusions

Our results deliver new insights about potential POI in southern Germany. The findings improve the geo-spatial understanding of tick ecology and disease transmission and aid public health professionals, general practitioners, and the public in recognising potential risk areas and reinforcing prevention efforts. A high degree of spatial heterogeneity was observed both at the district and sub-district level. To the best of our knowledge, sub-district spatial distribution of POI has not been previously investigated for the whole of Bavaria and Baden-Wuerttemberg with this level of geographic precision. Based on our heatmap and predictive map, some areas have the potential to be classified as additional TBE foci in the future, if confirmed by field work. Examination of the selected environmental covariates in isolation returned inconclusive results, emphasising that the various potential environmental determinants for TBE infection risk are best considered in combination. Therefore, multivariable ecological niche modelling appears to be a valuable tool for predicting TBE risk. The present model serves as a basis for what can be performed to interpolate the probability of TBE infection risk to other unsampled sites. It may be further calibrated, and variable inclusion and instrumentalisation can be re-evaluated to enhance its predictive performance.

## Figures and Tables

**Figure 1 microorganisms-10-00643-f001:**
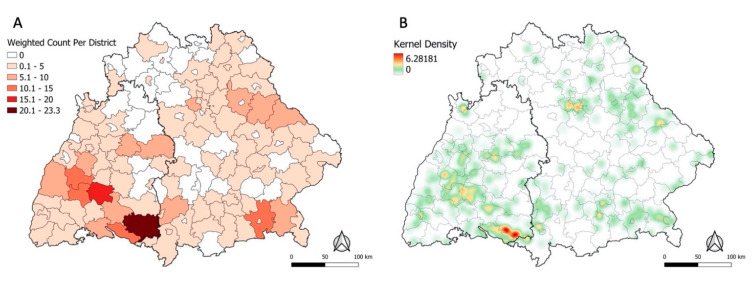
Sum of weighted self-reported POI per district from *n* = 359 cases from 2018–2020 (**A**). Heatmap of weighted self-reported POI from 2018–2020 with a bandwidth of 10 km (**B**).

**Figure 2 microorganisms-10-00643-f002:**
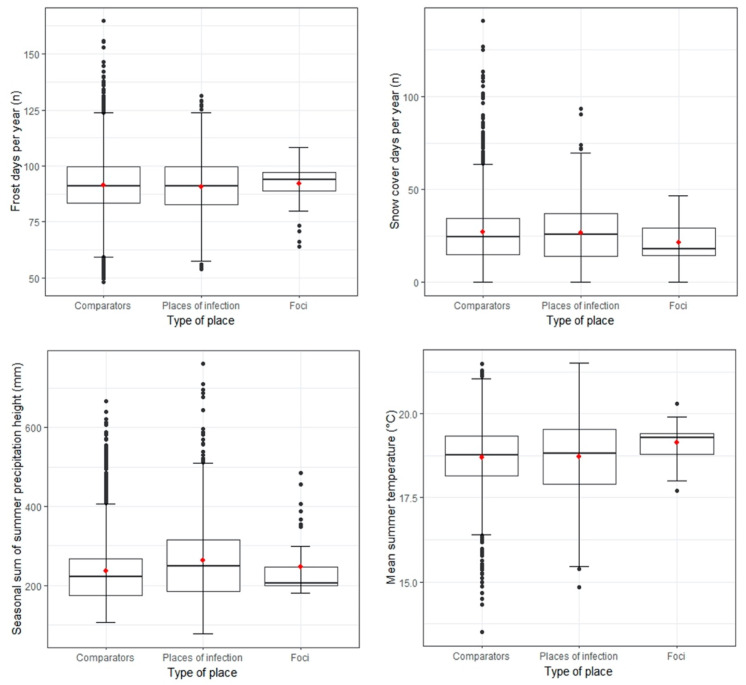
Distribution of selected covariates for comparator places (*n* = 1701), POI (*n* = 567), and natural TBE foci (*n* = 41) from 2018–2020; the red dots in the boxplots are the means. Environmental characteristics of further ecological covariates after variable inflation factor (VIF) selection are displayed in Appendix A.

**Figure 3 microorganisms-10-00643-f003:**
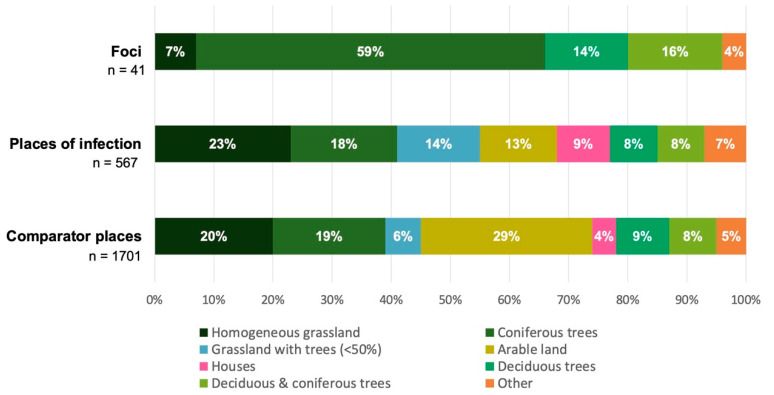
Proportion of land cover types for self-reported places of TBE infection with assumed TBE virus presence, natural TBE foci with confirmed TBE virus presence, and random comparator places as pseudo absences.

**Figure 4 microorganisms-10-00643-f004:**
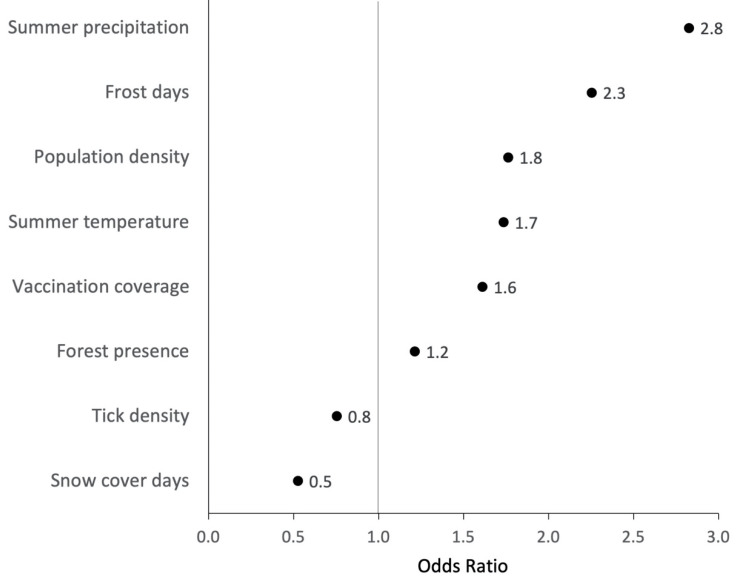
Direction and magnitude of effect of each covariate in determining the probability of the presence of a POI; comparing the 25% to the 75% percentile of each model covariate while keeping all other variables at the median value; raw odds ratios (ORs) and 95% confidence intervals (95% CI) are displayed in Appendix A.

**Figure 5 microorganisms-10-00643-f005:**
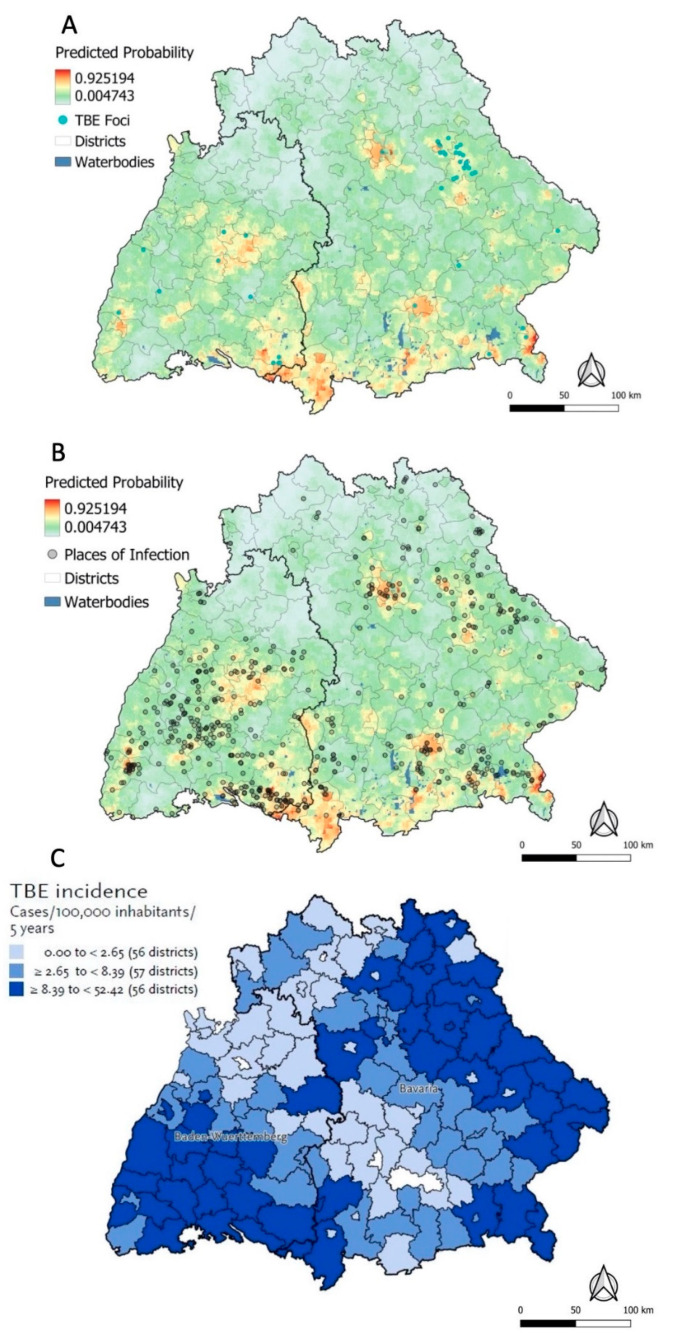
Predictive map for the probability of a place of TBE infection being present based on the ecological niche model superimposed with confirmed TBE foci (**A**); same predictive map superimposed with POI (**B**); excerpt of the RKI TBE risk map for 2020 based on incidence data from 2016–2020 (Adapted from Ref. [7], tertiles in the legend apply to all districts at risk in Germany) (**C**).

**Table 1 microorganisms-10-00643-t001:** Descriptive statistics of participant reports on probable places of TBE infection (POI).

**Type of Place Indicated**	**Study Participants (*n* = 359)**
Indicated points only	163 (45%)
Indicated polygons only	128 (36%)
Indicated both	68 (19%)
**Number of Places Indicated**	**Study Participants (*n* = 359)**
Indicated 1 place	235 (66%)
Indicated 2–5 places	119 (33%)
Indicated 6 or more places	5 (1%)
**Tick Bite yes or no**	**Study Participants (*n* = 359)**
None	120 (33%)
One or more	239 (67%)

**Table 2 microorganisms-10-00643-t002:** Size of self-reported TBE infection points and polygons, confirmed natural TBE foci, and comparator points and polygons (km^2^).

	Buffered Points (*n* = 320)	Polygons (*n* = 247)	TBE Foci (*n* = 41)	Comparator Points (*n* = 960)	Comparator Polygons (*n* = 741)
Min.	0.28	0.09	<0.01	0.28	2.30
Max.	0.50	5.34	0.09	0.50	2.30
Median	0.50	1.93	0.01	0.50	2.30
Mean	0.44	2.29	0.03	0.43	2.30
Standard Deviation	0.07	1.45	0.02	0.07	

## Data Availability

Restrictions apply to the availability of these data. Data were obtained through the Robert Koch Institute and the Department of Microbiology of the German Armed Forces and are available from the authors with the permission of the providing parties.

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
