# Peer review of "Geo-Spatial Characteristics of 567 Places of Tick-Borne Encephalitis Infection in Southern Germany, 2018–2020"

_microorganisms, 2022, doi:10.3390/microorganisms10030643_

Round 1
Reviewer 1 Report
Thank you for an interesting paper on TBE distribution. I have one question regarding the location data - given people were invited to the study, in some cases, three years after being diagnosed, how do you know the location data provided by participants is accurate - it is a long time ago to remember accurately?
Minor suggestions:
Line 16 add ‘of’ in front of ‘the’
Line 119 Reference needed for snowball sampling
Line 397 spelling error ‘Strengths’
Author Response
I have one question regarding the location data - given people were invited to the study, in some cases, three years after being diagnosed, how do you know the location data provided by participants is accurate - it is a long time ago to remember accurately?
Response: We recruited cases on an ongoing basis and always interviewed them as soon as we could, not after 3 years. The median delay between reported symptom onset and our reception of the POI marked on the maps was 83 days. This information was added in the manuscript (lines 87-89 and 205-207).
Minor suggestions:
Line 16 add ‘of’ in front of ‘the’
Response: This sentence is grammatically correct as it is
Line 119 Reference needed for snowball sampling
Response: Implemented
Line 397 spelling error ‘Strengths’
Response: Corrected
Reviewer 2 Report
The authors present a very interesting on the geo-spatial characteristics of TBE infections in Bavaria and Baden-Wuerttemberg, Germany. It is the first spatially high-resolution modelling of human TBE cases based on confirmed TBE virus foci and a new study on self-reported places of infection.
Both the method and its scientific implementation are correct and the paper could well be published without changes. Nevertheless, I recommend the following minor improvements, which are listed below.
1) In the abstract one searches in vain for the "geo-spatial characteristics" of the places of TBE infections, i.e. which predictors characterize the places. There is also no reference to the use of 41 natural TBE foci. Those readers who are not interested in the technical specifications of the spatial interpolation (compilation of a risk map) want to know exactly that.
2) I think the mapping of the known TBE foci of Bayern and BW is a highlight of this paper. Unfortunately, these are difficult to see in FIG. 5B. I therefore propose to depict these in Fig. 5A and remove them from Fig. 5B. In addition, Fig. 6A could be represented as Fig. 5C, which makes the comparison easier. For this purpose, Fig. 5C should be redrawn to the same scale as the other maps (not a map of Germany, just Bavaria and BW). The data on this is available at RKI. Fig. 6 is then no longer needed and a comparison is much easier for the reader.
3) In order to be able to compare the maps better, the border between Bavaria and BW should be entered in all maps (as in Fig. 5).
4) Fig. S2 is not required because the paper only considers the spatial distribution of the TBE cases.
5) It would certainly be of interest to see maps of the predictor altitude, population, summer precipitation, frost days and land cover (forest type) in the supplements. The reader wants to check visually whether these are correlated. For example, there should be more frost days at higher altitudes and summer precipitation is higher on the northern edge of the Alps. Unfortunately, no correlations (scatter plots) between the predictors and the POIs were given in the supplements. Many readers will look for these.
6) Line 96: "... 2 millimetres of inaccuracy ...". It is unusual to specify the accuracy of a printed map in mm, as these can now be zoomed on the computer. Delete that part of the sentence.
7) Line 276: "... at a 0.5 cut-off point ...". Often a cut-off threshold of 0.5 is used to distinguish between predicted presence and absence. Why wasn't the cut-off threshold optimized, e.g. by applying the sensitivity-specificity sum maximization approach (Youden Index). This should at least be discussed. A binary map for predicted presence/absence superimposed with all places of infection and TBE foci (Fig. 5B) is missing. Why actually? This is a key result of the study.
8) Fig. 3. In general, it is not made easy for the reader to quickly interpret the synonyms used. If I interpret Fig. 3 correctly, foci = confirmed TBEV presence, place of infection = assumed (self-reported) TBEV presence, and comparator places = pseudo TBEV absence. This should be added to the caption so that the graphic is self-explanatory.
I really enjoy the discussion and look forward to the final paper.
Author Response
1) In the abstract one searches in vain for the "geo-spatial characteristics" of the places of TBE infections, i.e. which predictors characterize the places. There is also no reference to the use of 41 natural TBE foci. Those readers who are not interested in the technical specifications of the spatial interpolation (compilation of a risk map) want to know exactly that.
Response: Wording adapted
2) I think the mapping of the known TBE foci of Bayern and BW is a highlight of this paper. Unfortunately, these are difficult to see in FIG. 5B. I therefore propose to depict these in Fig. 5A and remove them from Fig. 5B. In addition, Fig. 6A could be represented as Fig. 5C, which makes the comparison easier. For this purpose, Fig. 5C should be redrawn to the same scale as the other maps (not a map of Germany, just Bavaria and BW). The data on this is available at RKI. Fig. 6 is then no longer needed and a comparison is much easier for the reader.
Response: Good idea! Changed it accordingly.
3) In order to be able to compare the maps better, the border between Bavaria and BW should be entered in all maps (as in Fig. 5).
Response: Implemented
4) Fig. S2 is not required because the paper only considers the spatial distribution of the TBE cases.
Response: Figure deleted
5) It would certainly be of interest to see maps of the predictor altitude, population, summer precipitation, frost days and land cover (forest type) in the supplements. The reader wants to check visually whether these are correlated. For example, there should be more frost days at higher altitudes and summer precipitation is higher on the northern edge of the Alps. Unfortunately, no correlations (scatter plots) between the predictors and the POIs were given in the supplements. Many readers will look for these.
Response: Raster maps for the above-mentioned covariates can now be found in the supplementary materials. Regarding the suggested scatter plots, I consulted with the coauthors and we agree that the provided boxplots are sufficient and more relevant than scatterplots, given that POI are binary and not continuous. The correlation/direction & magnitude of effect between predictors and POI presence is brought about by the odds ratios in Figure 4.
6) Line 96: "... 2 millimetres of inaccuracy ...". It is unusual to specify the accuracy of a printed map in mm, as these can now be zoomed on the computer. Delete that part of the sentence.
Response: Deleted this part of the sentence
7) Line 276: "... at a 0.5 cut-off point ...". Often a cut-off threshold of 0.5 is used to distinguish between predicted presence and absence. Why wasn't the cut-off threshold optimized, e.g. by applying the sensitivity-specificity sum maximization approach (Youden Index). This should at least be discussed. A binary map for predicted presence/absence superimposed with all places of infection and TBE foci (Fig. 5B) is missing. Why actually? This is a key result of the study.
Response: Implemented --> applied sensitivity-specificity sum maximization approach and generated binary map based on optimised cutoff-threshold. Binary map can be found in the supplementary materials.
8) Fig. 3. In general, it is not made easy for the reader to quickly interpret the synonyms used. If I interpret Fig. 3 correctly, foci = confirmed TBEV presence, place of infection = assumed (self-reported) TBEV presence, and comparator places = pseudo TBEV absence. This should be added to the caption so that the graphic is self-explanatory.
Response: Implemented